# SeGS: Semantic-aware 3D Gaussian Splatting for Multi-turn Language-guided Robotic Grasping in Changeable Environment

## Abstract

In real-world applications, robots often need to perform multi-turn grasping, which requires them to accurately perceive and adapt to changeable environments. Current approaches extract complex semantics from 2D foundational models to implicitly represent 3D scenes. However, these methods necessitate a complete scene re-learning, leading to significant delays and inefficiencies in handling updated environments. To address this limitation, this paper introduces Semantic-aware 3D Gaussians Splatting (SeGS), a novel method that explicitly represents scenes with rich semantic information, enabling rapid scene updating in evolving environments. SeGS incorporates 3D Gaussian Splatting(3DGS) and integrates semantic features into each 3D Gaussian to capture contextual semantic details. By using explicit 3DGS and employing rend-and-compare strategy, SeGS allows for fast adaptation to scene changes, equipping robots to execute multi-turn grasping in changeable environments. Extensive experiments on continuous tasks demonstrate SeGS's ability to quickly reconstruct altered scenarios, facilitating swift task execution.

## 1 Introduction

This research investigates the task of multi-turn language-guided robotic manipulation with external intervention. This task has broad applications, such as housework, cooking, and various other daily-life scenarios Liu et al. (2024); Moriuchi & Murdy (2024). To effectively tackle these tasks, we identify two primary challenges: Firstly, the robot must comprehend the scene at a semantic level to accurately localize target objects. Secondly, the ongoing execution conducted by both humans and robots results in an evolving environment, requiring the robot to swiftly adapt to changes in the scene. For example, as the task of block building shown in Fig. 1, a robot moves blocks to target positions indicated by a language query with the intervention of human. In each round, as the position of a block changes, the robot is expected to promptly respond to the alteration and proceeds to execute the grasping action for the next round.

Recent efforts Shen et al. (2023); Rashid et al. (2023) only partially solve these challenges. Particularly, these methods Shen et al. (2023); Rashid et al. (2023) incorporate semantic information from pre-trained 2D vision-language models into Neural Radiance Fields (NeRF) Mildenhall et al. (2021) through multiple posed images. This results in semantic-aware feature fields of environments, enabling the localization of targets using language queries. However, these approaches often require retraining to reconstruct the entire scene, even with minor local changes, hindering their ability to respond promptly to environment updates and carry out continuous tasks. On the other hand, some other methods Xu et al. (2023); Tang et al. (2023) incorporate semantic information directly into explicit representations of point clouds from RGB-D cameras. Nevertheless, these representations rely on the external depth sensor for capturing the scene depth. These works motivate us to further address challenges related to *open-vocabulary scene understanding*, particularly in grounding and grasping target objects and *efficiently updating scenes.*

This paper presents a novel approach, Semantic-aware 3D Gaussian Splatting (SeGS), that rapidly represents scenes by integrating 3D Gaussian Splatting with semantic information. (1) To address the challenge of

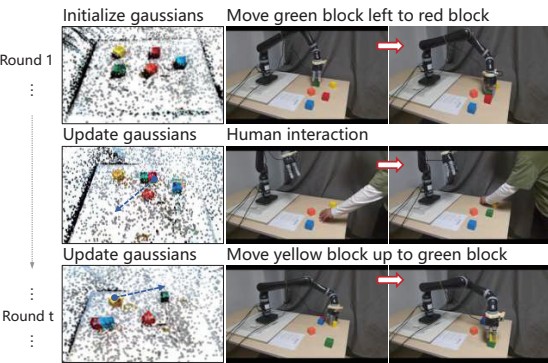

Figure 1: Illustration of the block building task. Our SeGS performs robotic manipulation continuously with the collaboration of human in a changeable environment. SeGS can quickly adapt to scene changes caused by either the robot itself or human, thus rapidly finishing a sequence of natural language commands.

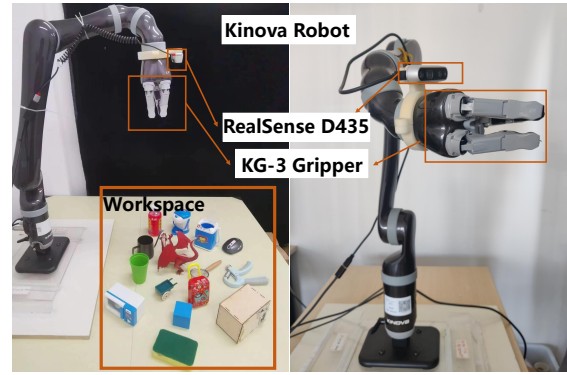

Figure 2: Our robotic grasping system and testing objects in our experiment.

*open-vocabulary scene understanding*, our method incorporates 3DGS and extend original 3DGS by distilling CLIP Radford et al. (2021) and DINO Caron et al. (2021) features, which enables precise object localization based on text descriptions. Specifically, we first capture multiple images of the initial scene and extract CLIP and DINO features. Then, the scene undergoes reconstruction, incorporating semantic information extraction. Upon receiving a text description, we use the CLIP feature to identify the relevant 3D Gaussians and locate the corresponding object-associated Gaussians using the DINO feature. Then dense point cloud of the identified object is generated and the point cloud is then forwarded to GraspNet Fang et al. (2020) for grasping pose generation. (2) To tackle the challenge of efficiently updating scenes, we propose a render-and-compare strategy that enables rapid optimization of the moved object's Gaussians. The main contributions in this paper are three-fold: 1) We propose SeGS, a novel system that integrates 3DGS with semantic information to facilitate multi-turn robotic grasping in changeable environments. 2) By leveraging CLIP and DINO features, SeGS extends original 3DGS and facilitates open-vocabulary scene understanding and accurate object localization from textual descriptions. 3) We develop render-and-compare strategy that enables rapid scene updates, reducing the need for full scene reconstruction.

## 2 Related work

### 2.1 Neural Radiance Fields (NeRF) in robotics

The NeRF Mildenhall et al. (2021) have witnessed significant advancements in recent years, revolutionizing the reconstruction of high-quality scenes from posed RGB images. This breakthrough has catalyzed the integration of neural fields into various robotics applications such as grasping Kerr et al. (2023); Ichnowski et al. (2021); Dai et al. (2023); Yen-Chen et al. (2022) and navigation Sucar et al. (2021); Zhu et al. (2022); Rosinol et al. (2023); Adamkiewicz et al. (2022), owing to their high reconstruction quality. Existing efforts predominantly focus on harnessing NeRF to reconstruct the entire scene. Unfortunately, due to the MLP-fitting nature of NeRF for scene properties, the process of capturing multiple images and training for novel scene reconstruction is especially time-consuming, limiting its applicability to only static object grasping. Evo-NeRF Kerr et al. (2023) attempted to accelerate scene updation to achieve sequential grasping, but its NeRF-based approach leads to the loss of object geometry information, thus inevitably requiring multiple perspective images, which is time-consuming and makes it hard to perform language-guided grasping. In contrast, our approach stands out by leveraging Gaussian Splatting Kerbl et al. (2023) for more efficient scene reconstruction and updation. This efficiency and fast speed are crucial for enabling our model to perform target-driven grasping, allowing it to grasp specific objects based on user-defined use cases in both

static and *changeable* scenes. To the best of our knowledge, we are the first to reconstruct scenes using only RGB images in *changeable* scenes for target object grounding and grasping.

## 2.2 Language-guided grasping

The fusion of Computer Vision (CV) and Natural Language Processing (NLP) enhances language comprehension for robots. Earlier studies Guadarrama et al. (2014); Hatori et al. (2018); Shridhar & Hsu (2018); Nguyen et al. (2020); Chen et al. (2021) achieved object grounding with 2D models. Recent approaches Cheang et al. (2022); Sun et al. (2023) combine visual grounding with 6D pose estimation Lin et al. (2022) for robotic grasping, yet they may lack precision in fine-grained object manipulation. Progress has been made in integrating semantic information into 3D representations, exemplified by works like CLIPort Shridhar et al. (2022) and PerAct Shridhar et al. (2023), which extract semantic information from point clouds or scene depth. Other methods optimize rich semantic information within the reconstruction of the 3D scene, with F3RM Shen et al. (2023), LERF-TOGO Rashid et al. (2023) using RGB images to train NeRF and GNfactor Ze et al. (2023) using RGB-D information to get the voxelized representation of the scene. Though RGB-D information enables the GNFactor to generate the scene representation at a high speed, it requires additional depth information and the alignment of features with voxels when the scene changes. While F3RM is closely related, our work uniquely focuses on Gaussian Splatting technology and employ render-and-compare strategy, enabling robots to adeptly conduct sequential operations in either isolated environments or with external intervention.

## 2.3 Gaussian Splatting

Gaussian Splatting Kerbl et al. (2023) is a recently proposed method that acquires 3D Gaussians by adapting density control during training, utilizing input photos from multiple viewpoints for shape fitting in 3D space. Given its efficient scene representation, it emerges as a natural choice for scene reconstruction in the SLAM field Keetha et al. (2023); Yugay et al. (2023); Yan et al. (2023). Besides, some paper has combined it with semantic features recently Qin et al. (2023), but they are mainly focused on static vision task. Despite Gaussian's gaining attention for its real-time rendering capabilities, this method has yet to be applied to robotic grasping and manipulation tasks, a gap that this paper aims to address.

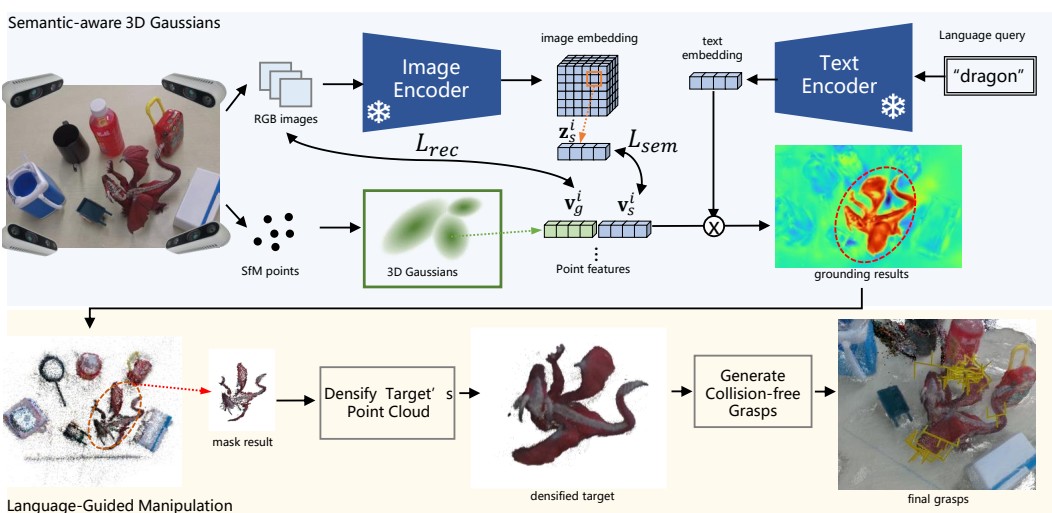

Figure 3: Pipeline of our proposed SeGS system. The camera is first utilized to capture multi-view RGB images and generate sparse SfM points. Then, semantic features are extracted and distilled to 3D Gaussian Splatting during scene reconstruction. Given a language query, we use distilled features to isolate the corresponding depicted object. After grounding the target object, we obtain the target point cloud by back-projecting the depth rendered from Gaussians belonging to region of target object. Finally, we generate the collision-free grasp poses from the located point cloud of target object.

# 3 Problem setup

Our goal is to develop a robot capable of efficiently executing multi-turn and interrelated commands with minimal latency, whether in isolated or interfered environments. The task can be conceptualized as manipulating building blocks, where both the robot and a human can pick up and place blocks, potentially stacking or combining them over multiple rounds to achieve the final goal.

To address this, we abstract the problem as one involving a fixed environment with several movable objects that can be stacked or concatenated. The system is initialized with a set of images $I_{init} = I_j^{init}$ and guided by language commands $C = \{c_i\}$. In real-world scenarios, the robot encounters two common situations: 1) Performing continuous operations independently; and 2) Handling and adapting to external intervention during the execution of multi-turn commands.

We refer to the former as an isolated environment and the latter as a interfered environment. In both settings, successful task execution requires robots to swiftly update scene representation $U = \{u_i\}$ from *few or even single image* $I_{upd} = \{I_{upd_i}\}$ and generating related 6-DoF grasp poses $\mathbf{T} = \{\mathbf{T_i} = (\mathbf{R_i}, \mathbf{t_i})\}$ and movement distance $d = \{\mathbf{d_i}\}$ for each sub-task, where $\mathbf{R_i}$ stands for the gripper's rotation matrix and $\mathbf{t_i}$ denotes the coordinates of the gripper's center. In essence, our method can be formed as maintaining a representation for the scene $Si = \mathbf{SeGS_{Upd}}(S_{i-1}, u_{i-1})$, while $S_0$ is initialized from $I_{init}$. With this representation, we can accomplish current task and subsequently update the scene, which can be achieved through $\mathbf{T_{i+1}}, \mathbf{d_{i+1}}, u_{i+1} = \mathbf{SeGS_{Grasp}}(S_i, c_{i+1}, I_{upd_{i+1}})$.

**Isolated Environment.** Even in an isolated environment where only the robot is executing tasks, it is impractical to pre-define all the $\mathbf{T_i}$ and $d_i$ due to potential errors during execution. For instance, if the gripper's position is not optimal when placing an object, it can lead to deviations from the intended outcome. Therefore, a updation strategy is required to ensure more precise positioning and grasping, enabling the robot to sustain multi-turn operations effectively.

**Interfered Environment.** We designed this task with the understanding that external interventions are inevitable in practice. During multi-turn grasping, instead of operating independently in a isolated environment, the robot should accurately estimate object movements with minimal latency, enabling it to swiftly adapt to changes in the surroundings

We assume the following conditions, which have been addressed in prior research. First, the robot is equipped with a calibrated wrist-mounted RGB camera. Second, objects are graspable by the robot arm. Third, the movable objects are positioned on a flat surface, such as a table.

# 4 Method

**Overview.** We give an overview of our SeGS, as in Fig.3: First, the initial scene is reconstructed (Sec. 4.1), and semantic features are distilled (Sec. 4.2). Second, target objects in the language query are grounded (Sec. 4.3). Finally, grasp poses are generated and the updated scene is managed (Sec. 4.4).

## 4.1 Preliminary: Gaussian Splatting

Given a set of images $I_{img}$, we use 3D Gaussian Splatting to reconstruct the scene, which uses a set of 3D Gaussian ellipsoids, denoted as $e = \{e_i\}$, to approximate objects in 3D space. Each Gaussian encompasses coordinate $\mathbf{p} \in \mathbb{R}^3$, a covariance matrix $\mathbf{\Sigma}$, color $c$ and opacity $o$. The first two parameters signify the shape and position of the Gaussian in 3D space, while the latter two represent color attributes. For rendering, 3D Gaussians are projected onto the image plane, and each pixel's color results from the alpha-blending of the 3D Gaussians covering it in the order of distance. In the training phase, loss is computed to optimize the parameters mentioned before. With this rasterization method, 3D Gaussian Splatting facilitates real-time rendering and rapid training, which proves beneficial for tasks sensitive to reconstruction and rendering speed. Importantly, Gaussian also offers a point-cloud based representation for the scene, thereby enhancing tasks such as grasping and enabling incremental operations.

## 4.2 Semantic feature distillation

To supervise the distillation process, we extract dense visual features aligned with open vocabulary language using the CLIP model. Addressing the limitation of CLIP, where features are originally extracted solely at the image level, previous studies, such as Shen et al. (2023), utilized the output from the penultimate layer of the model. This approach provides patch-level alignment between language and image. We denote the features of the patch $p_i$ of the image $I$ as $\mathbf{f}_{p_i}^{\mathrm{CLIP}} \in \mathbb{R}^{768}$ and the whole feature map of $I$ as $\mathbf{F}^{\mathrm{CLIP}}$. However, this method may not adequately capture object boundaries, potentially impeding further object grounding. Consequently, alongside $\mathbf{F}^{\mathrm{CLIP}}$, we also incorporate DINO features as DINO has demonstrated remarkable results for unlabeled data. These are denoted as $\mathbf{F}^{\mathrm{DINO}}$ for image $I$ and $\mathbf{f}_{p_i}^{\mathrm{DINO}}$ for the patch $p_i$ of image $I$.

Subsequently, we extend the Gaussian Splatting by embedding semantic features $\mathbf{f}_{\mathbf{e}_i}$ within each 3D Gaussian, where $\mathbf{f}_{e_i} = \mathbf{f}_{e_i}^{\mathrm{CLIP}} \oplus \mathbf{f}_{e_i}^{\mathrm{DINO}} \in \mathbb{R}^{768+384}$. We employ differential Gaussian rasterization to obtain dense semantic features for each 2D image $I$, denoted as $\hat{\mathbf{F}}^{sem} = \{\hat{\mathbf{f}}_i^{sem}\}$, where $i$ stands for each pixel of $I$ and $sem$ can refer to either CLIP or DINO features.

$$\hat{\mathbf{f}}_i^{sem} = \sum_{i \in N} \mathbf{f}_{e_i} \alpha_{e_i} \prod_{j=1}^{i-1} (1 - \alpha_{e_j}), \alpha_{e_i} = o_{e_i} * \sigma_{e_i} \tag{1}$$

where $\sigma_{e_i}$ is evaluated from projected 2D Gaussian with covariance $\mathbf{\Sigma}_{\mathbf{e}_i}$ and $o_{e_i}$ is the opacity of the 3D Gaussian which is also used in rendering RGB image, thus connecting the semantic features and RGB, leading to the semantic features be more spatially precise and be more fit with the 3D geometric shape of the object.

It is worth noting that, unlike NERF, 3DGS represents objects using explicit Gaussians. However, incorporating high-dimensional semantic features in each Gaussian would lead to significant memory overhead. A complex scene may involve over 100,000 Gaussians. Furthermore, during the backward phase, updating semantic features necessitates the computation of the term $\frac{\partial \hat{\mathbf{f}}_{p_i}^{sem}}{\partial \mathbf{f}_j}$ for each pixel and for all the Gaussians' semantic features covering it. Consequently, this term alone can incur a cost of $\Sigma_{i=0}^{P} num_i * (768 + 384)$, where $P$ denotes the image resolution and $num_i$ represents the 3D Gaussians covering it. Despite these challenges, we still aim to maintain the dimension of $\mathbf{f}_{e_i}^{\mathrm{CLIP}}$ to enable open-vocabulary understanding.

Instead, we adjust the resolution of rendered feature images during training. While a single feature image may be low-resolution, multi-view images complement each other, enhancing the precision of distilled features. As shown in the following experiment, the feature map rendered from distilled features is significantly more precise than one directly extracted from an image. At this stage, we have completed the extraction of feature maps and the extension of 3DGS.

While training, RGB and features are rendered using differential Gaussian rasterization in different resolutions. The loss function is composed of two part:

$$\mathcal{L} = \mathcal{L}_{rec} + \mathcal{L}_{sem}, \tag{2}$$
$$\mathcal{L}_{rec} = (1 - \lambda_1)\mathcal{L}_1 + \lambda_1 \cdot \mathcal{L}_{\mathrm{D-SSIM}}, \tag{3}$$
$$\mathcal{L}_{sem} = \mathcal{L}_{sem}^{\mathrm{CLIP}} + \lambda_2 \cdot \mathcal{L}_I^{\mathrm{DINO}} \tag{4}$$
$$= \mathcal{L}_2(\mathbf{F}^{\mathrm{CLIP}}, \hat{\mathbf{F}}^{\mathrm{CLIP}}) + \lambda_2 \cdot \mathcal{L}_2(\mathbf{F}^{\mathrm{DINO}}, \hat{\mathbf{F}}^{\mathrm{DINO}}) \tag{5}$$

where $\mathcal{L}_1$ denotes the L1 loss between the rendered image and the ground truth image; $\mathcal{L}_{\mathrm{D-SSIM}}$ is the SSIM loss between them. We have $\mathcal{L}_2$ for the L2 loss between the rendered semantic feature and the ground truth semantic feature. $\lambda_1$ and $\lambda_2$ serve as the regularization coefficients.

## 4.3 Object grounding

In this step, we aim to segment the object corresponding to the language query from the scene after scene reconstruction and feature distillation. Our grounding method involves the following steps. First, we apply the RANSAC (Random Sample Consensus) algorithm to isolate the points representing the table. RANSAC

is an iterative method for estimating parameters from observed data that includes outliers. Pre-excluding Gaussians representing the table expedites subsequent searches. Moreover, since the table is stationary, this operation is performed only once.

Second, we utilize CLIP model to encode the given query and calculate the cosine similarity with $\{\mathbf{f}_{e_i}^{\mathrm{CLIP}}\}$. We also use negative labels to filter out Gaussians to further accelerate the following process. The Gaussian with the highest similarity is chosen as the initial point, denoted as $p_{init}$. Third, we use BFS(Breadth-First Search)-alike algorithm to segment the full object. Beginning from $p_{init}$, we regard points with the cosine similarity of $\{\mathbf{f}_{e_i}^{\mathrm{DINO}}\}$ greater than a certain threshold and the distance lower than a certain threshold as its neighboring points. By iteratively performing this process, we can identify all the Gaussians constituting an object.

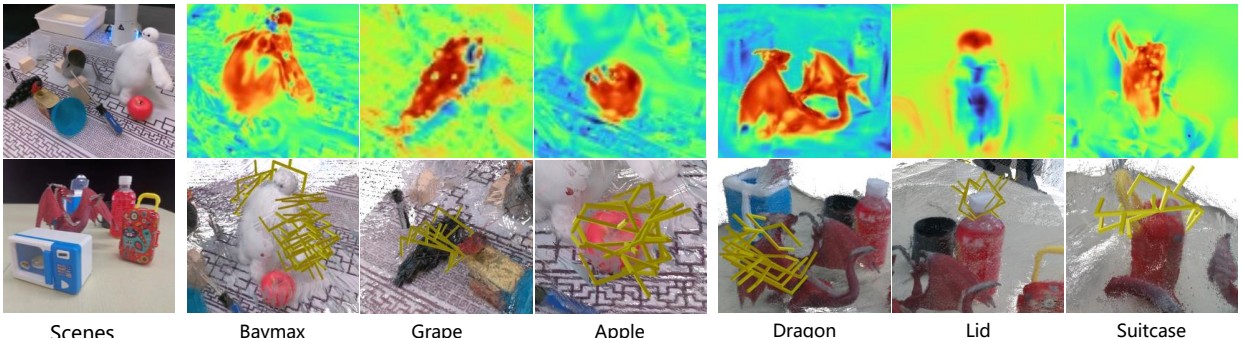

Figure 4: Qualitative results of our SeGS method. We show the generated heatmap given different language query corresponding to the target object (top row). Our method can generate grasp poses based on the point cloud from Gaussians (bottom row).

## 4.4   Robotic manipulation

**Basic Grasping.** Given a language query, we generate 6-DoF grasping poses for the corresponding object. First, we obtain the object's Gaussians, denoted as $\{p_i^{obj}\}$, using the method from the previous section. Next, we employ GraspNet, which takes a dense point cloud as input and outputs grasp poses for all objects on the desk. In our task, we create a dense point cloud $\{P_i^{obj}\}$ by applying differential Gaussian rasterization to depth and RGB images captured from multiple viewpoints based on $\{p_i^{obj}\}$. It is important to note that the depth information is an intermediate result of this process rather than data from a depth sensor. Following this, $\{P_i^{obj}\}$ serves as input for the network to generate corresponding grasps $\{G_i\}$. Notably, $\{P_i^{obj}\}$ only includes the point cloud of the target object, disregarding the surrounding environment, as if it were floating in space. Consequently, we perform collision detection against the original point clouds of the complete scene—represented by the centers of the Gaussians—to filter out gripper positions in $\{G_i\}$ that may collide with other objects. By directly utilizing the point cloud, we avoid the voxelization typically associated with NeRF-based methods, enabling a more detailed representation of elongated objects. This approach allows us to achieve zero-shot grasping of open vocabulary objects.

**Isolated Environment.** Since only a robot is involved in complementing the task, the key step is efficiently updating the scene after each object movement. Practically, only single-view image is available for this purpose. Our method leverages the editable nature of 3DGS and mainly follows a coarse-to-fine strategy.

Upon receiving an input command, we use a large language model (LLM) to identify the names of the relevant objects. Combined with the earlier described object grounding technique, we obtain the Gaussians for these objects, allowing us to determine their starting and ending positions.

Subsequently, in the real world, we physically grasp the object and place it in the designated location. Meanwhile, 3DGS related to the target are translated by the calculated distance. Till now, we finish roughly updating the scene. With just a modest amount of training on scene reconstruction using few or even single images after object movements in the real world, satisfactory results such as PSNR can be achieved in reconstruction, thus facilitating preparation for the subsequent grasping task.

For example, given a command: "put the pen on top of the sponge". LLM identifies the moving object as "pen", while referencing the "sponge". Consequently, it generates a language query:" pen and sponge". Through object grounding, the starting position $(x_1, y_1, z_1)$ and ending position $(x_2, y_2, z_2)$ for grasping are determined. Then, the pen is relocated, and the associated 3D Gaussians are shifted to the corresponding position simultaneously. After that, a single-view image is captured and utilized to refine the updated 3D Gaussians. With this, a grasping cycle concludes, and the robot is poised to execute the next command.

**Interfered Environment.** In the interfered environment, the translation and rotation of the object is unknown. Here, we design a method using a single-view image, denoted as $I_{mov}$ to predict the translation $\mathbf{d_{mov}} = [\Delta x, \Delta y, \Delta z]$ as well as the rotation $\mathbf{q_{mov}}$ represented by quaternion, which serve as the optimization parameters in this setting. To be specific, we initially employ the distilled semantic features to identify 3DGS corresponding to the language query, denoted as $\{e_i\}$. Then, during the optimization process, we translate $\{e_i\}$ by $\mathbf{d_{mov}}$, and rotate the orientation of them by $\mathbf{q_{mov}}$, while keeping other Gaussians still. Subsequently, two images are rendered: one with partial Gaussian $\{e_i\}$ denoted as $I_{obj}$, and the other with full 3D Gaussians denoted as $I_{mov,pred}$.

We define our loss function $\mathcal{L}$ as follows:

$$\mathcal{L} = \mathcal{L}_{fg} + \lambda_3 \mathcal{L}_{obj} \tag{6}$$

$$\mathcal{L}_{fg} = (1 - \lambda_1)\mathcal{L}_1(I_{mov}, I_{mov,pred}) + \lambda_1 \mathcal{L}_{\mathrm{D-SSIM}}(I_{mov}, I_{mov,pred}) \tag{7}$$

$$\mathcal{L}_{obj} = \mathcal{L}_2(d_{pred}, d_{gt}) \tag{8}$$

where $d_{pred}$ is the predicted location of the cup on $I_{obj}$; and $d_{gt}$ signifies the location of cup obtained through the output of the penultimate layer of the CLIP model on $I_{mov}$. Given that only seven parameters need optimization, the optimization process is rapid, typically completed in approximately 50 milliseconds. Subsequently we employ the grasping method mentioned earlier to generate 6-DoF grasping poses and implement the task. Our method stands apart from others by solely utilizing RGB images, thereby reducing hardware requirements.

## 5 Experiment

### 5.1 Environment and setup

In our physical robotic experiment, the system is deployed on a KINOVA Gen2 robot equipped with a 6-DoF curved wrist and a KG-3 gripper. An RGB camera mounted on the robot's arm captures RGB streams of the scene. The system operates on a desktop with an NVIDIA GTX 2080 GPU. The real robot and related devices used during the experiment are shown in Fig. 2.

### 5.2 Reconstruction and semantic distillation

In this chapter, we mainly compare our method with F3RM Shen et al. (2023) and LERF Rashid et al. (2023) in terms of scene reconstruction and semantic distillation using the dataset from F3RM Shen et al. (2023). The chapter's results are trained and tested on RTX4090. *Reconstruction Results.* In that we are dealing with robot tasks, we hope that the scene reconstruction process is as fast as possible, and also note that the "Training time" in the table includes the time to extract features from the RGB images and distill them to the target model. Besides, we use PSNR, SSIM, LPIPS as metrics for reconstruction quality. Specifically, PSNR measures signal quality by comparing it to a reference image. SSIM evaluates image similarity based on luminance, contrast, and structure. LPIPS is a learned metric using a neural network to measure perceptual image similarity. We compared our model with F3RM Shen et al. (2023) and LERF Rashid et al. (2023). The results are shown in Table 1. It can be seen that our method has faster reconstruction speed and higher reconstruction quality. With only 428 seconds, the PSNR metric still reaches 27.35.

**Semantic distillation results.** As for the semantic distillation, we mainly use the generated grasps as a measurement in the later chapter for practical consideration. Here we offer some qualitative results. During the experiment, we found that LERF requires the extraction of multi-scale semantic features, resulting in a longer processing time, with about 2 minutes for 48 pictures. Instead, F3RM is fast in doing so, but it failed to identify elongated objects such as whisk, whose heatmap is shown in Fig. 9.

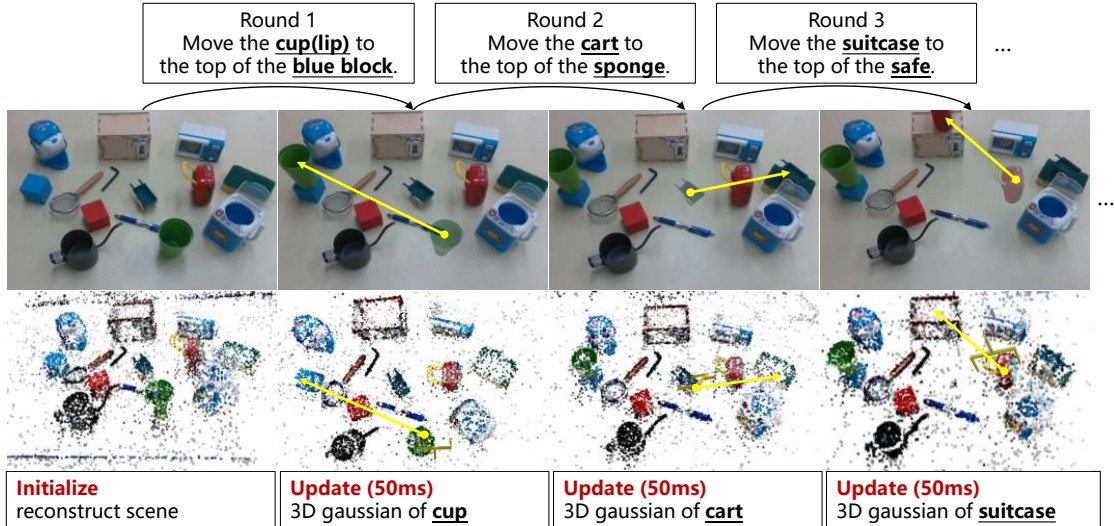

Figure 5: Example of testing scene Pictures in the first row compose one of the scenes used in testing. For this scene, we input three commands in turn. After the robot completes each command, we only give the robot a limited time to update the scene. Pictures in the second row are corresponding centers of the Gaussians and grasping poses.

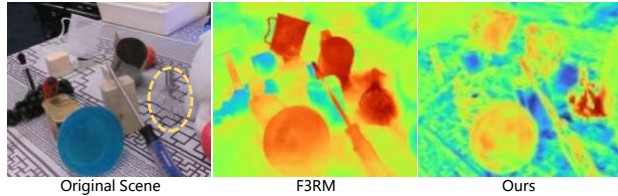

Figure 6: Qualitative results of the heatmap generated by our and baseline method. Given the language prompt "whisk", our method assigns a higher value to the heatmap region corresponding to the whisk compared to other objects, which facilitates further localizing the target objects.

## 5.3 Grasping accuracy in static environment

In this section, we compare our basic grasping method with others on the dataset publicly available from F3RM as well as in real world. The Table. 2 is the comparison of valid grasping poses' rates. Visualization of our method's grasping results is as Fig.4.

The failure of F3RM to grasp the whisk is due to two reasons: First, as can be seen from the previous text, the semantic information distilled was confused, resulting in the inability to effectively distinguish between the whisk and the cup, thus unable to determine the position of the whisk accurately. Another reason is that NERF provides implicit representation, and to generate the 6D pose, it needs to be voxelized. However, due to the fixed size of the voxel, there is a loss of precision in the voxelization process, which leads to the neglection of elongated objects. The screwdriver can still ensure a relatively high accuracy because the identification of the whisk is mainly depend on the elongated iron wire at its front, while the identification of the screwdriver is through its relatively thick handle, which is retained during voxelization.

Table 1: Results of rendering quality and training time.

| Model | PSNR ↑ | SSIM ↑ | LPIPS ↓ | Training Time(s)↓ |
|---|---|---|---|---|
| F3RM Shen et al. (2023) | 16.51 | 0.38 | 0.35 | 622 |
| LERF Rashid et al. (2023) | 24.45 | 0.78 | 0.32 | 1932 |
| Ours | **27.35** | **0.85** | **0.27** | **428** |

Table 2: Results of success rate by using estimated grasping poses

| Model | Whisk | Baymax | Suitcase | Dragon | Microwave oven | Blue Screwdriver |
|---|---|---|---|---|---|---|
| F3RM | 1/10 | **8/10** | 8/10 | **7/10** | **7/10** | **10/10** |
| LERF-TOGO | 5/10 | **8/10** | 7/10 | 6/10 | 5/10 | 9/10 |
| Ours | **7/10** | **8/10** | **9/10** | **7/10** | 6/10 | 8/10 |

## 5.4 Isolated changeable environment

In this section, we conducted experiments on a real machine. We first took photos from 30 different angles to reconstruct the initial scene and distill semantic information. Then, we sequentially input three commands for the robotic arm to grasp the object and place it in a specified position which is generated by LLM. One of the scenes used in testing is shown in Fig. 5, and the execution of our robot is shown in Fig. 7. As the setting mentioned before, we used a single-view image to update the scene each time, considering practical applications. Furthermore, we aimed to achieve as precise a reconstruction of the scene as possible within a limited time, which, in our experiment, is 50ms. For measurement, we focused on the moving objects, measuring the Euclidean distance between the reconstructed object position and the actual position. For our own method, we experimented with whether solely optimizing 3DGS or incorporating movement corresponding to the objects, denoted as Ours(w/o moving) and Ours(w moving) separately, and compared them with F3RM Shen et al. (2023) and LERF Rashid et al. (2023). Results are shown in Table. 3.

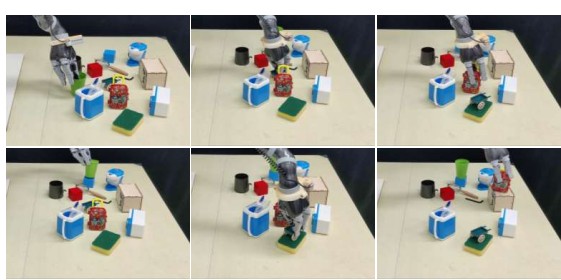

Figure 7: Qualitative results of updating 3D Gaussian. When robots moving target object by language query, corresponding 3D Gaussian are updated by the distance calculated from a priori Gaussian scene and the RGB images of current scene.

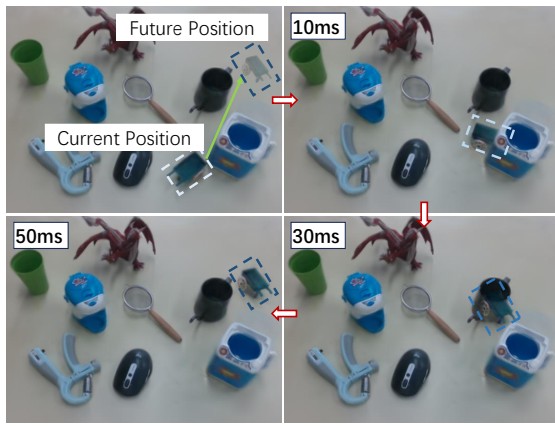

Figure 8: Optimization process of our proposed algorithm. The top-left image is the current scene, and the position pointed by the arrow indicates the future place of the cart. The remaining three images show the intermediate outcome during process of optimization.

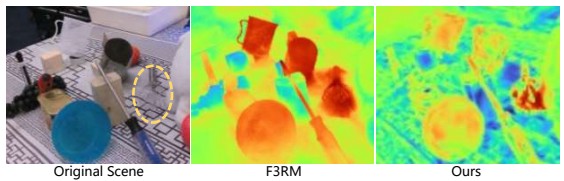

Figure 9: Qualitative results of the heatmap generated by our and baseline method. Given the language prompt "whisk", our method assigns a higher value to the heatmap region corresponding to the whisk compared to other objects, which facilitates further localizing the target objects.

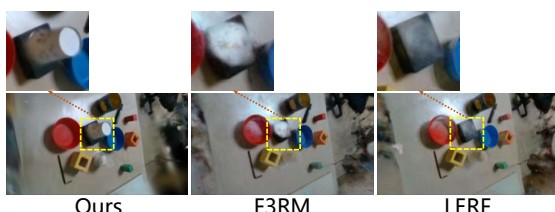

Figure 10: Comparison of reconstruction results by different methods. Both F3RM and LERF fail to reconstruct the scene after the robot moves the cup, thereby halting the pipeline of further tasks.

Table 3: The Euclidean Distance between the Predicted and GroundTruth After Object Movement

| Methods | Cup | Bowl | Suitcase | Dragon | Cart |
|---|---|---|---|---|---|
| F3RM Shen et al. (2023) | 0.1834 | 0.2432 | 0.1504 | 0.1785 | 0.1142 |
| LERF Rashid et al. (2023) | 0.1950 | 0.2207 | 0.1672 | 0.1679 | 0.1131 |
| Ours(w moving) | **0.0286** | **0.01315** | **0.0121** | **0.0107** | **0.0967** |
| Ours(w/o moving) | 0.1305 | 0.1923 | 0.1485 | 0.1688 | 0.1027 |

Table 4: Result of Grasping accuracy After Arbitrary Movement

| Methods | Time(ms) | Scene1 | Scene2 | Scene3 |
|---|---|---|---|---|
| F3RM | AnyTime | 0/5 | 0/5 | 0/5 |
| LERF-TOGO | AnyTime | 0/5 | 0/5 | 0/5 |
| | 10 | 0/5 | 0/5 | 0/5 |
| Ours | 30 | 2/5 | 3/5 | 2/5 |
| | 50 | **4/5** | **5/5** | **5/5** |

Table 5: Result of Grasping accuracy Using different Loss

| Loss | Time(ms) | Scene1 | Scene2 | Scene3 |
|---|---|---|---|---|
| w/o $\mathcal{L}_{fg}$ | 30 | 1/5 | 1/5 | 1/5 |
| | 50 | 2/5 | 2/5 | 1/5 |
| w/o $\mathcal{L}_{obj}$ | 30 | 0/5 | 1/5 | 0/5 |
| | 50 | 1/5 | 1/5 | 0/5 |

From the Table. 3, it can be seen that both LERF and F3RM have certain errors. Even with sufficient training time, as is shown in Fig. 10, due to the limited perspectives, these methods still cannot reconstruct the updated scenes.

## 5.5 Interfered environment

In this section, we validate that our method can accomplish weak real-time tasks even without using depth information. We place a few objects on a table and then randomly move an object. We aim for our method to achieve close to real-time updating. This part of the experiment also took place on a real machine environment. The experimental process is as follows: similar to the previous section, we first took photos from 30 different angles to initialize the scene and distill semantic information. Then, after moving a random object to a random place, we use a single viewpoint to capture a photo, predict the moving distance and direction, and generate grasp poses for the moved object. Grasping accuracy is measured. The intermediate optimization outcome of the scene4 is shown in Fig. 8. As we use the optimization method to predict the movement, there may exist some randomness, so that in Table. 4 we predict the location multiple times to test the accuracy.

From the Table. 4, it can be seen that F3RM and LERF-TOGO failed to do so as it fails to update the scene with only a single-view image available, while our method can effectively update the representation of the scene, thus generating grasps.

## 5.6 Comparison with explicit methods

Compared with explicit methods which use depth sensor, our methods stands out when dealing with transparent and metal objects. To validate this, we constructed scenes composed of metallic objects and transparent objects for the grasping experiment. We use grasping accuracy as metrics and results are as shown in Tab. 6.

Table 6: Result of grasping transparent and metal objects.

| Methods | Scene1 | Scene02 | Scene03 | Sum |
|---|---|---|---|---|
| Ours | 7/10 | 6/7 | 6/8 | 19/25 |
| Depth Sensor | 1/10 | 2/7 | 1/8 | 4/25 |

### 5.7 Ablation study

In this section, we mainly focus on the impact of different loss functions on the accuracy of object position movement in an interfered environment setting. We have tried the following three losses: 1) only $\mathcal{L}_{fg}$ 2) only $\mathcal{L}_{obj}$. We also use the grasping accuracy as measurement, as shown in Table 5. From the table and intermediate results, we can find that: in condition 1), without $\mathcal{L}_{obj}$, training is difficult to converge, and the success rate of grasping is very low. The reason for this is that the loss does not necessarily decrease as the object gets closer to its actual position. In fact, from the intermediate results, this is more like random drift. In condition 2), the grasping accuracy is higher than one. This is because with $\mathcal{L}_{obj}$, a rough estimation on movement can be ensured. However, without $\mathcal{L}_{fg}$, estimation is not precise and can lead to errors in extreme cases.

## 6 Conclusions and limitations

In conclusion, our method presents a promising solution for enabling robots to adapt to changeable environments, facilitating efficient multi-turn grasping with external intervention. Our Semantic Gaussian methodology can seamlessly extends to larger environments, such as room-scale settings, having the potential to conduct more sophisticated tasks, like room navigation. However, our method has some limitations. For example, it now only reconstructs a single large moving object, whereas future work will focus on enabling simultaneous reconstruction of multiple objects. Overall, our method demonstrates the potential for practical application in changeable robotic environments.

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

## A  Appendix

### A.1  Metallic and transparent objects

We design several scenes composed of metallic and transparent objects to validate that our method has advantage over others using depth sensor. One of the scenes is as bellow:

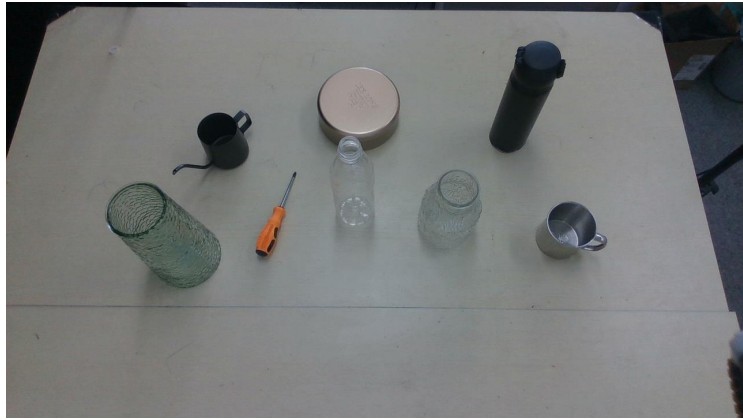

Figure 11: Metal and Transparent Objects

