# OpenReview forum: "SeGS: Semantic-aware 3D Gaussian Splatting for Multi-turn Language-guided Robotic Grasping in Changeable Environment"
_TMLR — Rejected by TMLR_

### Review · Reviewer_urxk · 2025-03-08

**Summary Of Contributions:**

This work introduces "Semantic-aware 3D Gaussians Splatting (SeGS)," a framework that improves perception for multi-turn grasping in changing environments. The main contributions include:

1. An explicit representation that embeds semantic feature from DINO and CLIP into 3D Gaussians.
2. A "render-and-compare" strategy that enables rapid adaptation to the environment changes for the scene representation.
3. Extensive experiments showing effectiveness of the approach in multi-turn grasping tasks.

**Audience:**

No

**Broader Impact Concerns:**

I have no concerns.

**Claims And Evidence:**

Yes

**Requested Changes:**

1. Conduct a more comprehensive literature review, especially focusing on recent works on Gaussian Splatting for manipulation.  (Critical to securing your recommendation)
2. Compare with recent Gaussian Splatting for manipulation approaches to demonstrate the advanced nature/superiority of this work. (Critical to securing your recommendation)
3. The quality of Figure 3 is not of high standard. I suggest using more consistent formatting and standardizing the capitalization of first letters. (Simply strengthen the work)

**Strengths And Weaknesses:**

Strengths:
1. This work studied an important problem: multi-turn language-conditioned grasp in changeable environments, which has broad applications.
2. The proposed approach effectively solves the target problem and enables rapid adaptation and robust grasp in the interfered environment.
3. The paper is easy to follow.

Weaknesses:
1. The literature review is not comprehensive and lacks closely related works [1] [2], which utilize Gaussian-based representation to enable real-time grasp sampling and adapation to changing environments.
2. Considering previous related works [1] [2], the novelty and contribution of this paper are limited.
3. Due to the neglect of recent related works, the baselines in experiments are not the most advanced approaches.
4. The adaptation method relies on language queries to handle object changes in interfered environments. However, during actual deployment, the other objects may be accidentally disturbed by the robot. The proposed adaptation approach appears insufficient to address this unintended interaction scenario.


[1] Ji, Mazeyu, et al. "GraspSplats: Efficient Manipulation with 3D Feature Splatting." 8th Annual Conference on Robot Learning.
[2] Zheng, Yuhang, et al. "GaussianGrasper: 3D Language Gaussian Splatting for Open-Vocabulary Robotic Grasping." IEEE Robotics Autom. Lett. (2024).

---

### Review · Reviewer_oQTF · 2025-03-08

**Summary Of Contributions:**

The paper introduces Semantic-aware 3D Gaussian Splatting (SeGS), a framework designed for multi-turn robotic grasping in dynamic environments. Unlike existing methods that often require time-consuming scene re-learning, SeGS explicitly represents scenes using 3D Gaussian Splatting (3DGS) with embedded semantic features, enabling rapid adaptation to environmental changes. The proposed render-and-compare strategy further accelerates scene reconstruction, facilitating swift task execution. Extensive experiments demonstrate SeGS’s effectiveness in dynamic settings, offering a practical solution for efficient and accurate robotic manipulation in real-world applications.

**Audience:**

Yes

**Claims And Evidence:**

Yes

**Requested Changes:**

- Abstract: Replace "implicit method" with "NeRF-based method" for clarity.
- Abstract: "rend-and-compare" -> "render-and-compare"
- Consistency in Notation: Ensure consistency in the use of punctuation in equations. For example, equations (3) and (5) use \times, while equations (6) and (7) do not. This should be standardized.

**Strengths And Weaknesses:**

### Strengths
1. **Efficient Scene Representation and Update:** SeGS leverages 3D Gaussian Splatting to represent scenes explicitly, enabling rapid updates to the scene representation. This is particularly advantageous in dynamic environments where objects frequently change positions.
2. **Open-Vocabulary Understanding:** SeGS supports open-vocabulary scene understanding, allowing the robot to localize and grasp objects based on natural language queries, even for objects not explicitly trained on.
3. **Real-World Validation:** The method is validated through real-world robotic experiments, demonstrating its practical utility in tasks such as block building and object manipulation. The supplementary video further supports these findings.

### Weaknesses
1. **Lack of Clarity on Render-and-Compare Strategy:**  Although the render-and-compare strategy is highlighted in the abstract, introduction, and Section 2.2, there is no detailed explanation of this strategy in the methodology section. Additionally, no ablation studies are provided to demonstrate its impact on performance.
2. **Limited Novelty in Combining 3DGS with Semantics:** The idea of combining 3DGS with semantic features is not entirely novel, as it has been explored in prior work [1]. The authors should discuss these related works in detail and provide experimental comparisons to highlight the advantages and disadvantages of SeGS.
- [1]. GraspSplats: Efficient Manipulation with 3D Feature Splatting (CoRL 2024)
3. **Insufficient Ablation Studies:** The ablation studies are somewhat lacking. More experiments are needed to fully understand the capabilities of SeGS:
- Impact of Feature Selection: What would be the effect of using only DINO or only CLIP features? How would this impact performance?
- Integration with Depth Information: Can SeGS incorporate depth information? Would this improve localization accuracy and speed? Additionally, how would using depth estimation models (instead of depth sensors) affect performance?
- Comparison with Other Explicit Representations: Besides NeRF-based methods, how does SeGS compare with point cloud, voxel, or point-based explicit representations in terms of speed and accuracy? What are the advantages of SeGS?
- Clarification on Depth Sensor Methods: In Section 5.6, the term "explicit methods which use depth sensor" is vague. The authors should specify which methods are being referred to and provide implementation details.
4. **Unsubstantiated Claims on Scalability:** The claim that "Our Semantic Gaussian methodology can seamlessly extend to larger environments, such as room-scale settings" is not supported by experiments. Given the significant memory overhead of 3DGS in large scenes, the authors should be cautious when making such statements.

---

### Review · Reviewer_U8EB · 2025-04-09

**Summary Of Contributions:**

This paper develops an approach for mult-turn robotic grasping that is amenable for dynamic scene variations. The approach builds on 3D Gaussian Splatting (3DGS) and enriches it with semantic features distilled from CLIP and DINO to produce a semantically meaningful, explicit 3D scene representation. The approach facilitates fast scene updates through a render-and-compare optimization strategy, allowing the system to adapt to scene changes without needing to retrain the entire model—a key limitation in prior NeRF-based approaches. The pipeline includes grounding language queries to 3D Gaussians, rendering semantic features, back-projecting them to point clouds, and then applying GraspNet to produce grasp poses. The system is demonstrated on real-world tasks involving both isolated and interfered (externally changed) environments.

**Audience:**

Yes

**Broader Impact Concerns:**

Not applicable.

**Claims And Evidence:**

Yes

**Requested Changes:**

Please refer to the weakness above, and kindly look at the following suggestions:

- Is it possible to provide evaluations on tasks beyond grasping? In most real-world robotic settings grasping is really only a first step, and so. it will be helpful to show that the approach can indeed be deployed in practical scenarios for manipulating objects

- Include comparisons with classical 3D vision or RGB-D pipelines (e.g., SAM+GraspNet, Mask R-CNN + depth heuristics), even if in simulation. This is important to understand the benefits comapred to using pretrained large-scale models.

- The use of language to ground objects is well-motivated but could be more thoroughly tested—e.g., how well does SeGS handle ambiguous, compound, or occluded queries? Some analysis of this will be helpful.

- please refer to the last point about potential grammar / semantic errors in the language descriptions and describe if they have affected the results in the paper. It will also be helpful to include *all* the task descriptions for clarity and transparency.

**Strengths And Weaknesses:**

Strengths

-The approach of combining Gaussian Splatting with semantic features from CLIP and DINO in the context of robotic grasping is novel to the best of my knowledge, resulting in a representation that is both rich in semantics and efficient to update.

-Fast Adaptation in Dynamic Environments: By avoiding full scene retraining, SeGS achieves real-time (∼50ms) updates, which is crucial for practical robotic manipulation in changing settings.

-Effective Object Grounding from Language: The method demonstrates open-vocabulary object grounding using language prompts—enabled by the semantic distillation of CLIP/DINO features into Gaussians.


-The system is evaluated across multiple axes—scene reconstruction quality (PSNR, SSIM, LPIPS), grasp success rate, update speed, and robustness to scene interference—showing consistent improvements over F3RM and LERF.


-The approach avoids reliance on depth sensors, making it easier to deploy on cost-constrained systems, and more easily deployable in diverse robot setups.


Weaknesses

1. Limited to Grasping as the Final Task: The method focuses solely on grasping as the robotic output. In practice, grasping is often the first step in a longer pipeline (e.g., manipulation, sorting, placing, tool use). Integrating SeGS into more complex downstream tasks would provide a stronger case for general utility.

2. Single Object Grounding & Manipulation: While effective, the approach is limited to moving one object at a time. Extending to scenes where multiple objects change simultaneously would be an important step toward real-world deployment. Further the approach is limited to only table-top manipulation and cannot be deployed in more generic settings.

3. Semantic Feature Memory Overhead: Although mitigated via low-resolution rendering and view fusion, embedding high-dimensional features into each 3D Gaussian leads to substantial memory and training cost, especially in large scenes.

4. Lack of Comparison to Non-NeRF Baselines: Comparisons are largely limited to NeRF-style methods (F3RM, LERF). It would be informative to include benchmarks against traditional vision-based or point-cloud based methods (e.g., using RGB-D + Faster-RCNN or SAM+GraspNet pipelines).

5. Limited Ablation on Language Understanding: The use of language to ground objects is well-motivated but could be more thoroughly tested—e.g., how well does SeGS handle ambiguous, compound, or occluded queries?

6. Potential grammar / semantic errors in the language descriptions. In the supplementary video, the language instructions seem to be vague or grammatically incorrect when compared with the robot executions. For example in 01:12 the instruction is "Move yellow block up to green block" and the video shows moving the yellow block "on top of" the green block. It is unclear what "up to green block" means? This should perhaps be rephrased to "on top of green block" I am not sure if these potential errors have affected the evaluations in the paper.

---

### Decision · Action_Editor_W9Es · 2025-05-27

**Recommendation:** Reject

**Comment:**

Reviewers appreciate the efficient scene representation (Reviewer U8EB oQTF), open-vocabulary understanding (Reviewer U8EB oQTF), evaluation of different aspects (Reviewer U8EB ), and paper writing (Reviewer urxk).
On the other hand, reviewers have concerns about the lack of discussion and comparison with related works (Reviewers oQTF and urxk about GraspSplats, GaussianGrasper), experimental comparison with related works, and ablation studies.
The concerns above cast doubt on individuals in TMLR's audience be interested in the paper.

**Audience:**

The paper lacks sufficient discussion of closely related works
The major concern raised by reviewers is about novelty when compared with existing works.

**Claims And Evidence:**

Reviewers appreciate the good results. The only concern is the advantage compared with existing works. Baselines used in the experiments are not the most advanced approaches, ablations and comparisons with related works are not strong, making the evidence not so convincing.